# Let's Make Ball Balancing Great Again: Why You Should Use Temporary Speed Reduction

**Gabriël Van De Velde [1,\*]**, **Björn Verrelst [2]**, **Dirk Lefeber [2]** and **Patrick Guillaume [1]**

[1]  Acoustics & Vibration Research Group (AVRG), Faculty of Mechanical Engineering,
    Vrije Universiteit Brussel, Pleinlaan 2, 1050 Brussels, Belgium; patrick.guillaume@vub.be
[2]  Robotics & Multibody Mechanics Research Group (R&MM), Faculty of Mechanical Engineering,
    Vrije Universiteit Brussel, 1050 Brussels, Belgium; bjorn.verrelst@vub.be (B.V.); Dirk.Lefeber@vub.be (D.L.)
\*   Correspondence: gavdevel@vub.be

**Abstract:** Automatic ball balancing is a technique adopted in rotordynamics to reduce unknown rotor unbalance automatically. This technique sounds appealing as it can ease a panoply of balancing issues considerably. The presence of stiction, however, scatters consistent qualitative balancing and led to a limited implementation in the industry. Temporary speed reduction, a recent technique, could be used as a countermeasure for the stiction-induced scattering. Presented in this paper is an in-depth study detailing how the technique should be implemented to guarantee effective balancing. By analysing a rotordynamic model of the Jeffcott kind, the influence of a multitude of parameters is studied such as the initial mass positions, the initial unbalance, the adopted speed profile, shaft damping, stiction and the speed reduction plateau of the adopted speed reduction strategy. The main findings of the study are that the adverse effects of stiction can be contained considerably using the speed reduction technique, especially in the under-excited range where a ball balancer behaves poorly when adopting a standard run-up profile. Finally, the speed plateau of the speed reduction technique should be selected carefully, preferably accounting for stiction, shaft damping and even more so the initial unbalance.

**Keywords:** ball balancer; unbalance reduction; temporary speed reduction; stiction

## 1. Introduction

Unbalance induced vibrations are an omnipresent issue in rotor dynamics. Several solutions have been provided in the past [1,2]. One technique to tackle these unbalance issues consists of using Automatic Ball Balancing. An Automatic Ball Balancer (ABB) comprises a circular retainer fixed on a rotating shaft holding a set of balls that can move concentrically around the shaft. Above the fundamental resonance frequency of the rotary system, this device can remove unbalance automatically.

An ABB is often applied when unbalance is varying. Washing machines, optical readers [3], angle grinders [4], sanding machines or lab centrifuges are typical cases where unbalance loads vary over time. However, if the unbalance does not vary, an ABB can still be beneficial considering manufacturing ease. For example, after final assembly, in-situ balancing might be required but not possible. In this case, Automatic ball balancing could tackle the unbalance issue and reduce manufacturing costs.

From an implementation point of view, an ABB should be a plug-and-play device that allows quick, safe and repetitive balancing. Unfortunately, the ABB is known to have a balancing capability of the statistical kind due to the presence of rolling friction [5–9] and becomes a plug-and-pray device. This major inconvenience led to a restricted implementation of the ABB in industry. A different issue occurs when nearing resonance as it induces violent bifurcative behaviour that can be described as of

the Sommerfeld Effect-kind [10–12]. Many research was devoted to analysing the conditions for the existence of these bifurcations [13–17].

In an attempt to parry the afore-stated unstable behaviour, countermeasures were provided in the past. For example, adding partitions in the raceway prohibits the balancing masses from swirling around the shaft when nearing resonance [17,18]. The addition of radial and or tangential springs to the balancing masses was considered as well [19].

As can be seen, the unstable bifurcative behaviour that occurs when nearing resonance can be obliterated with effective techniques. The list of practical solution regarding statistical scattering is however short. Ref. [7] states that "the ball balancer might not be suitable as balancing measure for appliances demanding very low residual unbalance. Reducing the friction and enlarging the driving force are the two major ways for further reduction of the residual unbalance". The use of multiple raceway tracks or balancing masses is a viable solution [20] as the addition of multiple down-scaled ball balancers have more freedom to relocate. Considering the one-ball ABB, a shock technique was presented in [21] whereby a sudden torque shock swirls the balancing mass to a different place. Another approach [22] consisted of using an observer to estimate the state of the balancer. Application of torque would allow relocating the sole balancing mass effectively. Unfortunately, the latter technique can only be applied to the one-ball ABB as the action influences all balancing balls identically.

Research [8] shows, by using Friction Maps, that a slow speed profile is advantageous as the ball relocation process only occurs near resonance. The use of a smooth raceway, thereby reducing friction is thus of the uttermost importance. However, this slow run-up strategy has a drawback as the bifurcative behaviour near resonance is detrimental and preferably avoided, by adopting a short run-up procedure. This contradiction and the added complexity of the proposed solutions regarding friction-induced scattering entailed a limited use of the ball balancer.

In a previous publication, we provided an effective means to tackle both of the mentioned issues above by introducing the Temporary Speed Reduction technique [9]. This minimalist approach works as follows: By quickly crossing the resonance frequency, unstable behaviour is limited. Reducing the rotational speed and nearing the resonance supercritically afterwards allows effective balancing safely and repetitively. The downside of this approach is the necessity of a controllable rotational speed.

The presented study aims to assess for a given rotary model whether the use of an ABB is auspicious, what its balancing capabilities are and if adopting the temporal speed reduction technique (TSR) is prolific. By answering these questions, we wish to facilitate and encourage the use of ball balancers. This will be done with an in-depth sensitivity analysis that accumulates the effect of multiple parameters to illustrate how friction-induced scattering is affected. Moreover, a sensitivity analysis regarding the TSR technique is presented.

The paper opens by recalling in Section 2 the working principle of an ABB. In Section 3, we derive a dynamic model of a Jeffcott rotor comprising an ABB with rolling friction. Section 3.2 will detail the TSR technique. Section 4 presents the balancing quality of the ball balancer depending on the initial ball pose, the initial rotor unbalance, the adopted speed profile, stiction and the effect of rotor damping. These effects will be detailed, adopting a standard run-up profile as well as its TSR counterpart.

## 2. Basic Concept of an ABB

This section starts by detailing the fundamental of an ABB with one balancing mass. By mounting this ABB on a Jeffcott rotor [1] the balancing capability of an ABB is then declared.

### 2.1. Fundamental

A simplified model is sketched in Figure 1 on which a point I representing a balancing ball with mass m is allowed to move freely in a raceway at given distance $r$ around D, the centre of a rotating shaft bound to rotate around origin O at given distance $z$ and angular velocity $\Omega$. The angular position between $z$ and $r$ is denoted by $\gamma_i$.

As the point mass I orbits origin O with revolution speed $\Omega$ and distance $l$, it experiences a centrifugal force $F_{c,i}$ following $\vec{OI}$. Being bound to its circumferential path around D, a reaction force impedes any radial movement with respect to D. As tangential movement is allowed, a residual force $F_{t,i}$ drives point mass I towards a higher total excentricity $l$. This force equates

$$F_{t,i} = mz\Omega^2 sin(\gamma_i). \tag{1}$$

One can see that the point mass I will tend to move in the same direction as the local deflection $z$, towards E ($\gamma_i = 0$, $l = z + r$). Equation (1) can be seen as the fundamental of the ABB: A single ABB will tend to add unbalance in the direction of the deflection it perceives.

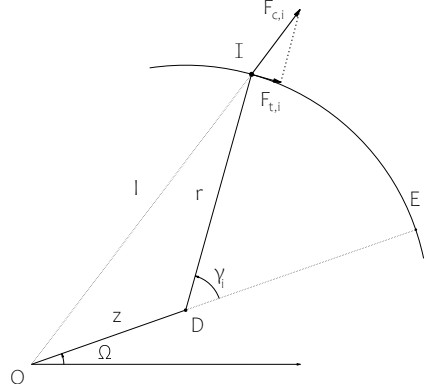

**Figure 1.** Fundamental sketch of a one-ball balancer with emphasis on forces acting on balancing mass I.

## 2.2. Automatic Balancing

The working mechanism of an automatic ball balancing device relies on the rotor it is built upon as the balancer only reacts to the local deflection that is perceived. This will be demonstrated using a Jeffcott rotor.

The state of the Jeffcott rotor is speed-dependent and can be divided into two main categories: the subcritical and the supercritical state. The angular position of the rotor deflection with respect to unbalance excitation in these categories is respectively 0° and 180°. Thus in subcritical regions, the balancer will amplify the initial unbalance, while at the supercritical state it will counteract this initial unbalance, giving an ABB its balancing capabilities at supercritical speeds.

As the phase shift of 180° occurs gradually around the critical speed, a third state arises, which is of the unstable kind. The unstable phenomenon originates when the phase lag is near 90°. As the balancing mass follows, the lagging eccentricity, its movement shifts the total unbalance, so that the response of the shaft drifts. This drifting leads to unstable transients and high intermittent vibration levels. As these bifurcative effects are out of scope for this study, we refer to the literature for more information on the matter.

By adding more balancing masses, it is possible to remove unbalance effectively, as any unbalance leads to shaft excentricity soliciting the balancing behaviour of each balancing mass.

The efficacy of the ABB relies on a well-defined phasal position of the shaft it is built-upon. Moreover, any friction inside the raceway potentially degrades the balancing capabilities of the ball balancer as nearing perfect balancing conditions weakens the forces acting on the balancing balls considerably. More specifically, the vibrational level $z$ and the correct ball position $\gamma_i$ both decrease the balancing forces considerably as can be seen in Equation (1). The detrimental effects of friction and how to mitigate them will be illustrated in Section 4.

### 3. Definition of the Rotordynamic Model

In this section, the equations of motion of a Jeffcott rotor and a concentrically mounted ABB with Coulomb rolling friction will be derived using the Lagrangian approach, provided with nominal parameter values. Then the TSR technique will be briefly introduced. Finally, the standardised speed profile used in the simulations will be presented.

#### 3.1. Equations of Motion

Using a relative cartesian base, we describe a concentrically mounted ABB (with $n$ balancing masses) on a Jeffcott rotor, as shown in Figure 2. In what follows, the non-deflected position of the shaft will be denoted by $O$, while the geometrical centre point of the shaft D. The centre of mass of the shaft G is positioned eccentrically with respect to D at a distance $\epsilon$, while ball $B_i$ is constrained concentrically at a set distance $R$ with respect to $D$. The absolute reference frame $(X, Y)$ has its origin located in $O$. The relative reference frame $(X', Y')$ has the same origin but uses the direction of the unbalance $DG$ as reference. $\psi$ denotes the imposed rotation between both frames.

The displacement of the shaft r is defined in the frame $(X'Y')$ as $(r'_x, r'_y)$. The relative angular position of the $i^{th}$ ball i.e., $\angle GDB_i$ is labelled $\phi_i$. Gravity is acting in the negative Y-direction. The equations of motion will be derived accounting for gravity for the sake of completeness, but it will be omitted further.

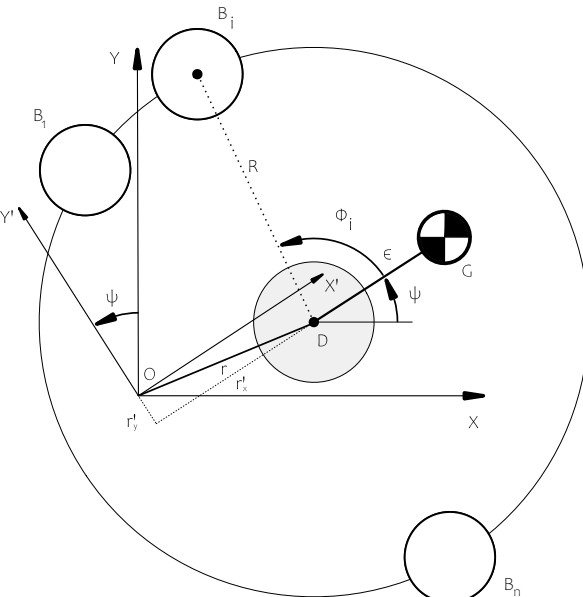

**Figure 2.** Coordinate definition for the Jeffcott rotor with a concentrically mounted ball balancer.

The vector of generalized coordinates **q** can then be defined as

$$\mathbf{q} = \begin{bmatrix} r'_x \\ r'_y \\ \phi_i \end{bmatrix} \quad i = 1, ..., n \tag{2}$$

Using these coordinates, the position of the shaft and the $i^{th}$ ball can be expressed as

$$\mathbf{r'}_{OG} = [r'_x + \epsilon, r'_y] \tag{3}$$

and

$$\mathbf{r'}_{OB_i} = [r'_x + R\cos(\phi_i), r'_y + R\sin(\phi_i)]i = 1, ..., n. \tag{4}$$

The characteristics of the Jeffcott rotor are defined by $k,c$ and $M$ which are respectively the perceived rotor stiffness, the apparent rotor damping and the rotor mass. There are $n$ balancing balls with mass $m_i$, the ball-raceway interaction is modelled as rolling coulomb friction by $\mu$ while the stabilising effect of the oil in the raceway is modelled as viscous ball damping by $D_i$. Finally, the total mass of the system, i.e., the shaft and the balancing masses is labelled $M_t$. The kinetic energy $T$ and potential energy $V$ of the system are

$$
\begin{aligned}
T = {} & \frac{M_t \dot{\psi}^2 r'^2}{2} + \frac{M_t \dot{r}'^2}{2} + M_t \dot{\psi}(r'_x \dot{r}'_y - r'_y \dot{r}'_x) + \frac{M\epsilon^2 \dot{\psi}^2}{2} + M\epsilon\dot{\psi}[r'_x \dot{\psi} + \dot{r}'_y] + \sum_{i=1}^{n} \frac{m_i R^2}{2}(\dot{\psi} + \dot{\phi}_i)^2 \\
& + \sum_{i=1}^{n} m_i R \dot{\psi}(\dot{\psi} + \dot{\phi}_i)[r'_y \sin\phi_i + r'_x \cos\phi_i] + \sum_{i=1}^{n} m_i R(\dot{\psi} + \dot{\phi}_i)[-\dot{r}'_x \sin\phi_i + \dot{r}'_y \cos\phi_i]
\end{aligned}
\tag{5}
$$

$$
V = \tfrac{k}{2}(r'^2_x + r'^2_y) + Mg[r'_x \sin(\psi) + r'_y \cos(\psi) + \epsilon \sin(\psi)] + \sum_{i=1}^{n} m_i g[r'_x \sin(\psi) + r'_y \cos(\psi) + R\sin(\psi + \phi_i)]
\tag{6}
$$

A Rayleigh dissipation function $F$ can be expressed as

$$
F = \frac{c}{2}[\dot{\psi}^2(r'^2_x + r'^2_y) + \dot{r}'^2_x + \dot{r}'^2_y + 2\dot{\psi}(r'_x \dot{r}'_y - r'_y \dot{r}'_x)] + \frac{D_i}{2}\sum_{i=1}^{n} \dot{\phi}_i^2
\tag{7}
$$

Using the kinetic and potential energies of the system $T$ and $V$ along with the Rayleigh dissipation function $F$ it is possible to determine the equations of motion following the Lagrangian principle, which yields:

$$
\begin{aligned}
M_t \ddot{r}'_x = {} & M_t \ddot{\psi} r'_y + 2M_t \dot{\psi}\dot{r}'_y + M_t \dot{\psi}^2 r'_x + M\epsilon\dot{\psi}^2 - kr'_x - c(\dot{r}'_x - r'_y \dot{\psi}) - M_t g \sin\psi \\
& + \sum_{i=1}^{n} m_i R \sin(\phi_i)(\ddot{\psi} + \ddot{\phi}_i) + \sum_{i=1}^{n} m_i R(\dot{\psi} + \dot{\phi}_i)^2 \cos(\phi_i)
\end{aligned}
\tag{8}
$$

$$
\begin{aligned}
M_t \ddot{r}'_y = {} & -M_t \ddot{\psi} r'_x - 2M_t \dot{\psi}\dot{r}'_x + M_t \dot{\psi}^2 r'_y - M\epsilon\ddot{\psi} - kr'_y - c(\dot{r}'_y + r'_x \dot{\psi}) - M_t g \cos\psi \\
& - \sum_{i=1}^{n} m_i R \cos(\phi_i)(\ddot{\psi} + \ddot{\phi}_i) + \sum_{i=1}^{n} m_i R(\dot{\psi} + \dot{\phi}_i)^2 \sin(\phi_i)
\end{aligned}
\tag{9}
$$

$$
\begin{aligned}
m_i R^2 \ddot{\phi}_i = {} & -m_i R \ddot{\psi}(r'_x \cos\phi_i + r'_y \sin\phi_i) - 2m_i R \dot{\psi}(r'_x \cos\phi_i + r'_y \sin\phi_i) \\
& + m_i R \dot{\psi}^2(-r'_x \sin\phi_i + r'_y \cos\phi_i) - D_i \dot{\phi}_i - m_i g R \cos(\psi + \phi_i) - F_{f,i} R\, i = 1, ..., n, \\
& - m_i R^2 \ddot{\psi} + m_i R \sin(\phi_i)\ddot{r}'_x - m_i R \cos(\phi_i)\ddot{r}'_y
\end{aligned}
\tag{10}
$$

with $F_{f,i}$ in Equation (10) denoting the Coulomb friction force impeding the tangent movement of the $i^{th}$ ball.

Friction is modelled as Coloumbic with a hyperbolic function, ensuring numerical stability. It is modelled as

$$
F_{f,i} = \mu F_n \tanh\left(\frac{\dot{\phi}_i}{v_{sat}}\right),
$$

with $F_{f,i}$ and $F_n$ respectively the resulting tangent frictive force impeding the tangent movement and the normal force acting on the $i$-th ball and $\mu$ the Coulomb friction interface value. Care should be taken when simulating friction forces holding a load over time [23]. In this case, $v_{sat}$, a sliding speed constant, has been fit as a trade-off between solver speed and simulation accuracy. A robust value has been set for $v_{sat} = 10^{-6} \frac{rad}{s}$.

The nominal parameter values used in this study are shown in Table 1 and are based on prior experimental research. More information can be found in [9].

**Table 1.** Nominal parameter values of the Automatic Ball Balancer (ABB)—Jeffcott rotor model.

| Parameter | Value | Parameter | Value |
|---|---|---|---|
| $M$ | 16.53 kg | $m_i$ | 7.6 g |
| $k$ | 5.8 kN mm$^{-1}$ | $R$ | 35 mm |
| $c_{ref}$ | 1.510 kN m s$^{-1}$ | $n$ | 2 |
| $|\epsilon|$ | 19.3 μm [1] | $\mu_{ref}$ | 0.002 |
| $\angle\epsilon$ | 0 rad | $D_i$ | 2.7 N mm s rad$^{-1}$ |

[1] G20 Following ISO.

### 3.2. Temporary Speed Reduction

As was mentioned previously, the presence of rolling friction in the ball-raceway interaction deteriorates the balancing qualities of an ABB. In previous research, we presented a technique called Temporary Speed Reduction (TSR) that allows us to mitigate the detrimental effects of rolling friction [9]. This research showed that a temporary speed reduction towards resonance frequency allows the balancing masses of an ABB to relocate towards a preferred position. This is possible as the Jeffcott rotor becomes sensitive to unbalance near its resonance frequency. This increase of vibration amplifies the balancing forces acting on the balancing masses (as seen in Equation (1)) and subsequently overcome friction. The balancing masses can thus relocate for proper unbalance compensation. The use of TSR allows us to significantly reduce unbalance in a quick, sound and replicable manner.

A standardised test speed profile has been defined encompassing TSR. A single test that lasts 90 s is detailed in Figure 3 and consists of 2 parts. The first part (0–20%) is a standard run-up. As this model mimics the real setup parametrised in Table 1, its regime speed of 167 Hz and resonance frequency of 94.2 Hz are used.

In the second phase (20–100%), the TSR technique is applied. Evaluating the balancing quality at 20% and 100% allows to assess the efficacy of the TSR technique as well. In the upcoming section, we will see that TSR is an efficient technique to contain frictive unbalance scattering, especially at low initial unbalance levels.

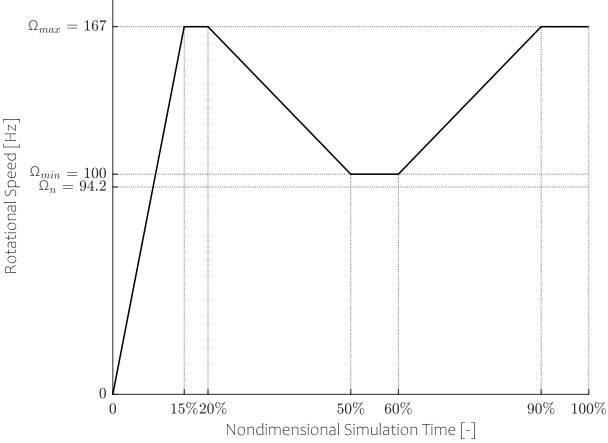

**Figure 3.** Standardized speed profile, nominal speed = 167 Hz. Total simulation time is 90 s.

## 4. Simulation Results

The studied parameters influencing the balancing quality of an ABB are

1. The starting position of the balancing masses;
2. The initial unbalance of the rotor;
3. The adopted speed profile;
4. Damping and raceway friction.

To preserve a coherent approach, the ISO balancing grade *G* as described by ISO 1940-11:2016 is used. This ISO standard defines rigid rotor balancing quality using a specific quality grade *G*. This quantity is defined as the product of the permissible residual shaft eccentricity (in [mm]) and the highest nominal operational speed of the machine (in [rad/s]), expressed in mm/s. The considered rotor should qualify the *G*16 balancing quality standard: the balancing quality is to be inferior or equal to *G*16. It is to be noted that the unbalance defined in Table 1 corresponds to *G*20.

Before performing the aforementioned sensitivity analysis, we will first look at a single test run.

### 4.1. Single Run

The selected configuration is defined in Table 1, with a given ball start configuration. This simulation will be compared to a reference run. The difference between both runs lies in the absence of the balancer.

The simulation is decomposed in Figure 4. Firstly a standard Amplitude graph compares the vibrational amplitude of the rotor with and without ABB. Then a Ball Position graph illustrates the movement of the balancing masses during the complete test run. The angular position shows the location of both balancing masses with respect to the initial unbalance while radial distance gives a perception of time. Finally the COG location graph (COG = Center Of Gravity) details the influence of ball movement on the residual unbalance location and thus balancing quality of the rotor. Hereby, the angular position denotes the location of the COG with respect to the initial unbalance while radial distance shows how severe the total amount of unbalance is (grade G).

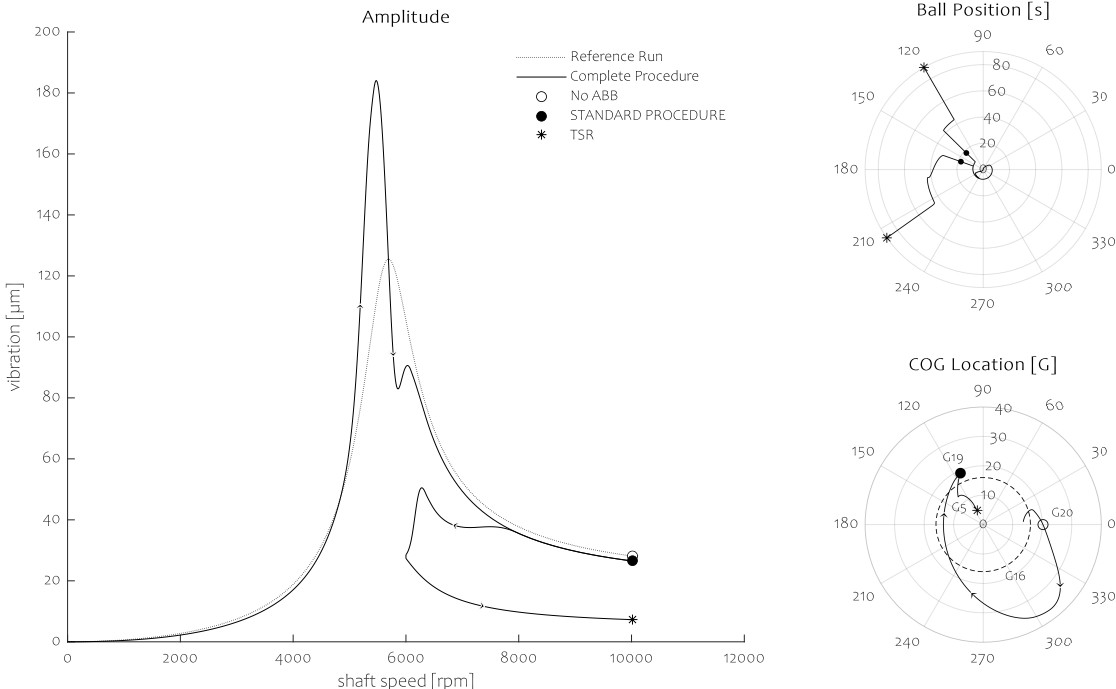

**Figure 4.** A single run-up following the aforementioned speed profile illustrating the temporal speed reduction (TSR) technique.

The Amplitude graph compares the vibration amplitude of the rotor with (continuous line) and without ABB (dashed line). It is noted that the sensitivity of the balancer unit to unbalance intensifies during resonance, slinging the balancing masses around as can be seen in the Ball position graph at an early stage. This relocation deteriorates the balancing grade of the shaft as pointed out in the COG Location graph. Nevertheless, the standard run-up procedure in this case leads to a balancing grade that varies considerably (*G*14 ⤚ *G*38 ⤚ *G*19). The balancing outcome (●) is less than the initial

unbalance of *G20* (○), although it still exceeds the balancing requirements (*G16*—dashed in the COG location graph). TSR mitigates this issue (∗) as illustrated in the Amplitude graph when comparing both techniques at nominal speed (10,000 rpm).

After the standard run-up procedure (●), the rotor speed is reduced. This reduction increases the sensitivity of the shaft to unresolved unbalance resulting in a vibration increase. The balancing forces acting on the balancer grow until the frictive thresholds are exceeded, releasing the balancing masses. The masses relocate to a configuration reducing the residual unbalance as can be seen in the Ball Position graph near $t = 40$ s. Increasing the rotational speed again desensitizes the balancer to residual unbalance so that both balancing masses remain fixed. Application of TSR (∗) in this case results in a considerable balancing quality (*G5*) thus qualifying for the *G16* standard.

### 4.2. Start Position of Balancing Masses

Ball positioning at the start of a run-up has a significant impact on final balance quality. In practice, their position cannot be chosen by any means. Their position is thus a variable that has to be taken into account. Therefore, we consider this variability as a first influence by moving the start position of each ball to 50 equispaced positions over the raceway. Since both balancing masses are identical, the complete analysis consists of 1275 simulations as nearly half of the computations are omitted.

Figure 5 presents the results of the analysis. In this Figure, the Probability Density Function (PDF) of both the standard (STD) and TSR procedures are displayed for an initial *G20* unbalance (○). There is an evident impact of the initial ball position, well-illustrating the non-repeatable aspect of rolling friction. It follows that the majority of balancing outcomes are *G15* for the standard approach (●) and *G11* for TSR (∗). Concerning the afore-stated ISO balancing standard of *G16*, both methods qualify in 85% and 80% of the cases for STD and TSR, respectively.

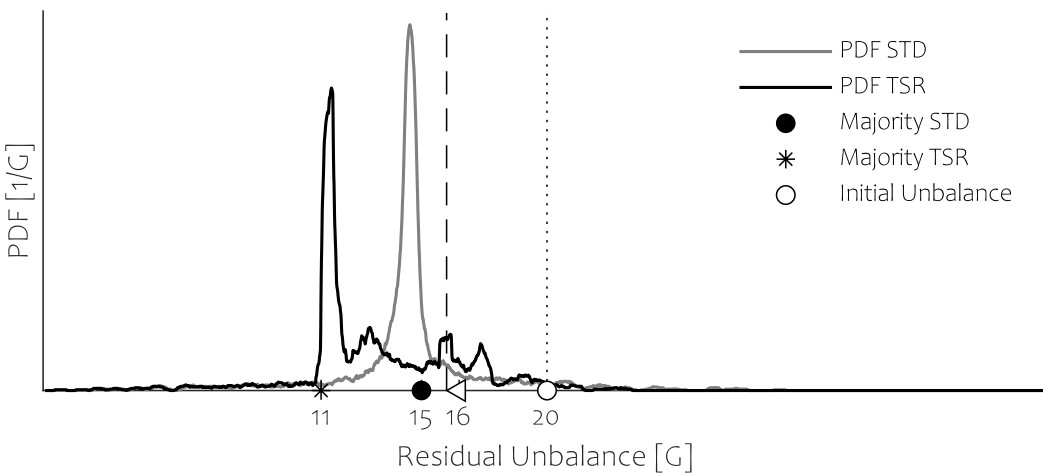

**Figure 5.** Probability density function of the residual unbalance for an initial unbalance G20 (dashed line), standard (grey) and TSR (black) procedure.

Summarizing all balancing outcomes with these values obfuscates the variability aspect as it is clear that in some cases, TSR deteriorates the STD balancing outcomes. Figure 6 illustrates these peculiar cases, showing each TSR outcome along with its STD counterpart. The graph is divided into several regions, as defined in Table 2, along with their respective occurrence rate.

Region I is divided into regions I.A (26%) and I.B (67.5%), depending on whether respectively TSR worsens or improves the STD outcome.

If the ball balancer is well-chosen for the Jeffcott rotor, all balancing outcomes will reside in I. Both procedures are thus able to limit unbalance within the initial boundary.

The success rate of the STD and TSR methods is respectively 95% and 98%. This means that in the majority of cases, both methods achieve a balancing class smaller than the initial *G20*. The analysis can

be concluded by stating that TSR consistently improves the STD approach for an initial unbalance of *G*20. Deciding whether the STD or TSR technique should be applied, however, depends on the adopted balancing standard. We restrain ourselves from making a suggestion, as the upcoming sections will alleviate this complex decision.

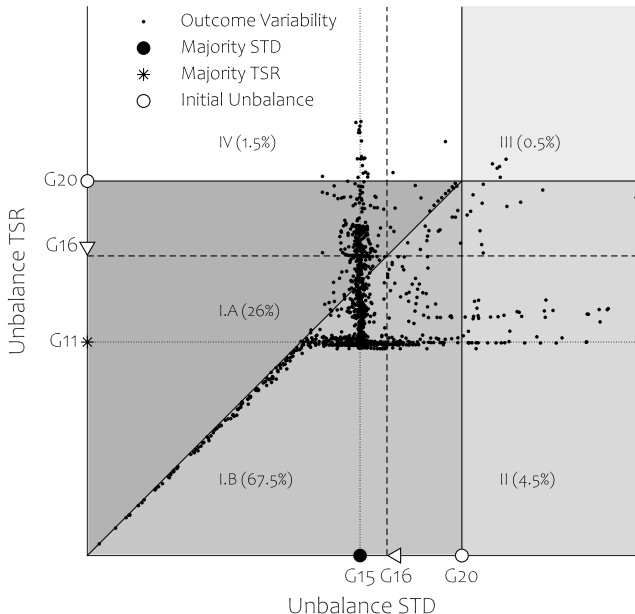

**Figure 6.** Comparison of the balancing outcome of the standard (STD) and TSR procedure.

Nevertheless, should the balancing outcome be not prolific, there is always the possibility to restart the appliance. Doing so, the balancing masses will have different start positions that will likely entail proper balancing in the next run. In the following subsection, a similar analysis will account for varying initial unbalance.

**Table 2.** Occurrence of each balancing outcome state.

| Region | State | Occurrence |
|---|---|---|
| *I.A* | $G_{STD} < G_{\mathbf{TSR}} \leqslant G_{INI}$ | 26.0% |
| *I.B* | $G_{\mathbf{TSR}} \leqslant G_{STD} \leqslant G_{INI}$ | 67.5% |
| *II* | $G_{\mathbf{TSR}} < G_{INI} < G_{STD}$ | 4.5% |
| *III* | $G_{INI} < \{G_{STD}, G_{\mathbf{TSR}}\}$ | 0.5% |
| *IV* | $G_{STD} \leqslant G_{INI} < G_{\mathbf{TSR}}$ | 1.5% |

*4.3. Initial Unbalance*

It is clear that by changing the initial ball configuration of the ABB, there is a pronounced scattering in balancing quality.

Another unknown is the initial unbalance that the system (without ABB) intrinsically has. The initial unbalance in a real system is statistically scattered as it depends on production, assembly and operating conditions. Therefore, the influence of the initial unbalance is to be studied next.

The influence of the initial unbalance, in conjunction with the scattering phenomenon illustrated in Section 4.2, leads to the data portrayed in Figure 7.

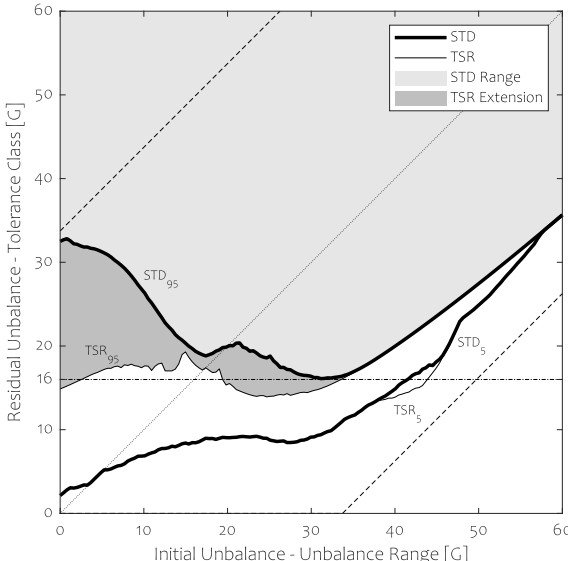

**Figure 7.** Lines: Influence of initial unbalance on residual unbalance following the standard configuration of Table 1 and afore-stated speed profile. Area: Valid initial unbalance range for a given tolerance class.

Figure 7 depicts the residual unbalance outcome of the balancer for a given initial unbalance. The central dotted line reflects the initial unbalance conditions: data below this reference line shows an improvement in balance quality, while data above shows balancing quality degradation. The black dashed lines mark the boundaries in between which the ABB can operate as they correspond to the total unbalance when both balancing masses are in- or out-of-phase with respect to the initial unbalance.

As the data scattering is highly non-normally distributed, average and variance are no robust measures of scale to quantify the statistical dispersion. A percentile analysis is thus preferred. In order to hint a sense of their distribution, the 5-percentile and 95-percentile isoline are depicted for both techniques. Doing so, a loss of information occurs. For example, a pronounced gathering of balancing outcomes between these two isolines would go unnoticed. Though, this presentation style is not erroneous as it still allows us to show the boundaries of the balancing behaviour. The 95 percentile isoline is therefore convenient as it allows us to quantify the global balancing quality.

We show that initial unbalance has an evident influence on the residual unbalance scattering. There is a noticeable gathering at higher initial unbalance towards the lower unbalance boundary for both procedures. At the lower initial unbalance side, more dispersion occurs: as there is few or no initial unbalance, the ABB has to cancel itself out. Due to rolling friction, the balls tend to scatter in a wide range as the driving forces (whose magnitude depends on the total unbalance) are limited. Better results can be achieved using the temporal speed reduction technique. Next, the 5-percentile isoline of both procedures is nearly identical, which is logical as TSR only influences high residual unbalance. The temporal speed reduction approach is an improvement, especially at reduced initial unbalance levels. In general, an important conclusion is to be made: an over-dimensioned ABB will balance poorly.

Figure 7 can alternatively be interpreted using the concept of balancing area: Given a balancing standard that has to be met (for example *G*16, horizontal dash-dotted line), a valid initial unbalance range can be defined for which 95% of the balancing outcomes are lower than the required balancing standard. Any initial unbalance in this range will lead to a residual unbalance meeting the balancing standard at least 95% of the time. By varying the balancing standard, the associated valid initial unbalance range changes as well. One can see that a balancing area or range can be defined. The STD

Range in Figure 7 shows a valid initial unbalance range for a given balancing standard. Balancing outcomes would preferably all reside on the lower operation boundary of the balancer as the complete unbalance operation range of the balancer would then meet any tolerance. Unfortunately, there is a considerable discrepancy between $STD_{95}$ and this lower boundary. The TSR extension improves this behaviour, particularly in the lower initial unbalance range. In the case of the balancing standard $G16$, one can see that there is no initial unbalance that would qualify the balancing standard using the STD procedure. By applying the TSR approach, a valid range occurs in $[G0 - G3, G19 - G34]$. It is possible to define efficiency by comparing this valid unbalance range and its theoretical maximum. This maximum unbalance range would occur if the $STD_{95}$ percentile isoline would lay on the lower operation boundary so that the valid unbalance range equates $[G0 - G50]$. Comparing the size of both ranges allows to state that the adopted speed reduction technique can guarantee proper operation of this specific balancer for the $G16$ tolerance class in 36% of all cases for nominal damping and stiction.

### 4.4. Effect of Speed Profile

The effect of the speed profile can be declared by studying the immobility of the balancing masses in Figure 8.

We define immobility as the ratio between the amount of all ball configurations that are not able to relocate at all and the total amount of ball configurations possible, for a given initial unbalance and rotating speed. This ratio is calculated with the balancing masses having no speed. In Figure 8, the initial unbalance is set as presented in Table 1. As a Jeffcott rotor is sensitive to unbalance near resonance, it is to be expected that the immobility drops, indicating that effective balancing can occur. The width and depth of this valley are influenced by damping and stiction with respect to the amount of initial unbalance as will be shown in Section 4.5. It is now clear that a steep speed profile prohibits effective balancing. By using the TSR technique, acceleration rate becomes less of an issue as immobility is lowered as long as needed, without compromising balancing stability. In this aspect, a gradually increasing speed profile as proposed in [8] seems effective, albeit omitted due to the violent intermittent vibrations occurring near resonance. This can be seen in Figure 9 whereby the 95% percentile isoline of the STD strategy using a $3x$ slower run-up is compared with the prior STD and TSR outcomes when adopting the regular speed profile. From this graph, it is clear that better balancing conditions can be attained and that a slower run-up seems advantageous. However, the unstable behaviour occurring near resonance slings the balancing masses, potentially aggravating their balancing outcome. This artefact can be perceived in the unbalance range $[G10, G21]$ whereby outcomes are even worse than their STD counterpart. The balancing inconsistency along with the prolonged detrimental operation near resonance favours steep profiles, resulting in a poor balancing quality. It is thus beneficial to adopt the TSR technique as it allows us to swiftly cross over resonance and balance effectively in a repetitive manner.

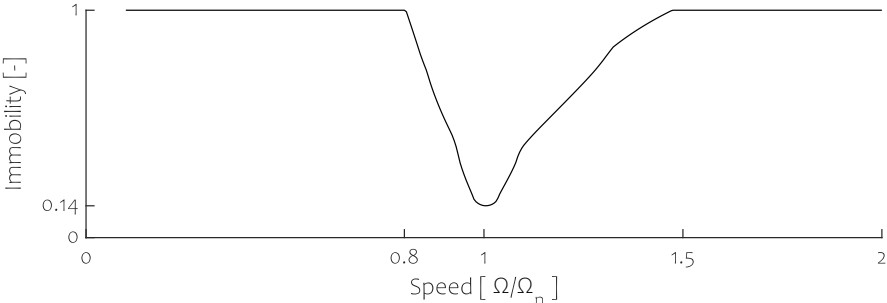

**Figure 8.** Immobility in function of rotational speed for an initial unbalance of $G20$.

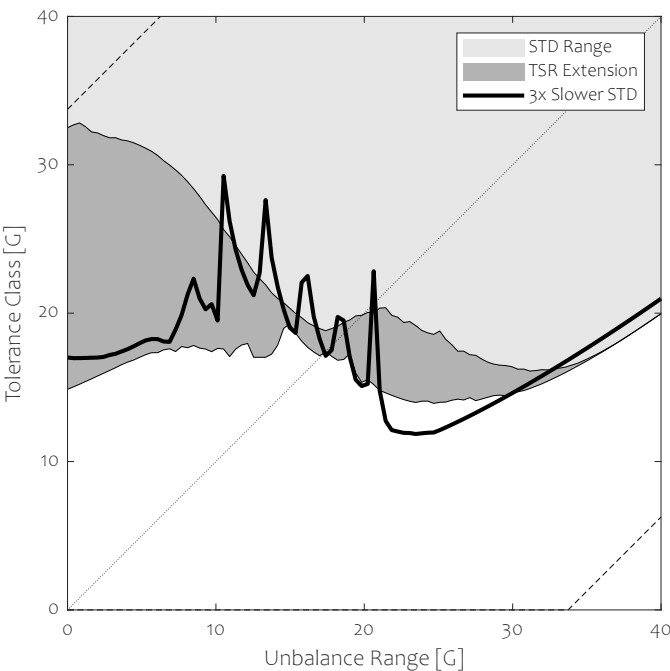

**Figure 9.** Influence of the speed profile on balancing range. A slower speed profile tends to mimic TSR. Intermittent vibrations when crossing resonance however annihilate this improvement.

### 4.5. Influence of Damping and Static Friction

We have seen that the initial ball starting positions and the initial unbalance level do influence the balancing behaviour of an ABB considerably. Other important parameters are the rolling friction and damping of the system. How these affect the global balancing quality will be investigated next by considering immobility.

In Figure 10, immobility is shown for a reference initial unbalance of *G*20 (as defined in Table 1). Reference values for stiction $\mu_{ref}$ and damping $c_{ref}$ are defined in Table 1 as well. These are scaled using the following amplification factors:

$$\begin{cases} c_r = \frac{c}{c_{ref}} = & \{0.5; 0.6; 1; 1.4\} \\ \mu_r = \frac{\mu}{\mu_{ref}} = & \{0.25; 0.5; 1; 2; 5\} \end{cases}$$

It is in this range that the impact of damping and stiction is analysed. It is to be expected that both the STD and TSR techniques are impacted, albeit the influence of both parameters is dissimilar. The areas depicted in Figure 10 correspond to a given stiction value. Darker shading denotes an increase of stiction thus immobility. The thickness of these areas indicates the contribution of damping. The central line in each area illustrates immobility for nominal damping ($c_r = 1$). The vertical dashed black lines denote the two characterising speeds of the adopted speed profile.

A detrimental aspect of increased damping is the reduction of stability in the mobility valley as the phase lag of the Jeffcott rotor occurs over a wider speed range. It is to be noted that this instability limit is influenced by the amount of initial unbalance and is most severe for limited initial unbalance values. A severe increase of damping in combination with restricted stiction will thus impact the lowest stable subcritical speed that can be attained. For the sake of simplicity, the lower speed plateau of the TSR procedure has been selected to guarantee stable and effective balancing reduction in the majority of cases.

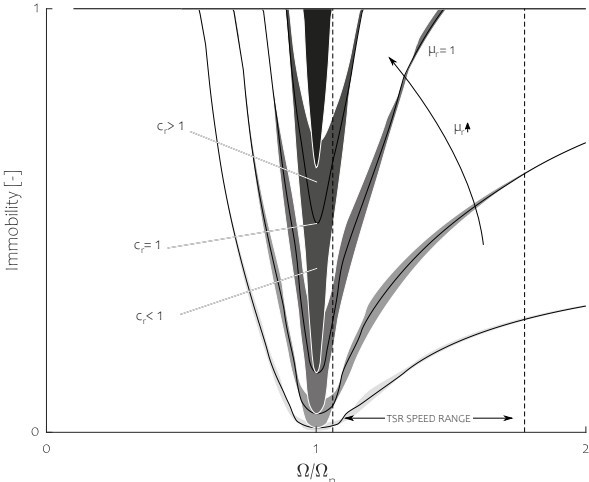

**Figure 10.** Influence of damping and rolling friction on balance immobility for a nominal initial unbalance of *G*20. Damping $c_r = \{0.5; 0.6; 1; 1.4\}$. Friction $\mu_r = \{0.25; 0.5; 1; 2; 5\}$.

Immobility is the lowest near resonance due to the vibration increase as dynamic stiffness is absent. Therefore damping has a significant impact on immobility near resonance. By increasing damping, the balancing capabilities of the balancer deteriorate as immobility increases drastically. In contrast, an excessive amount of stiction withstands any relocation attempt although some stiction facilitates stable balancing in the overcritical region.

The influence of stiction on balancing capability is shown in Figure 11a. The results for the STD approach are ambivalent. In the under-excited region, i.e., for an initial unbalance lower than the total ball unbalance *G*34, a sweep up occurs that limits the valid unbalance range for low stiction. This is because stability of operation is only ensured in a narrow upper speed segment of the adopted speed profile. Therefore in this initial unbalance range, the balancing outcomes are more spread than otherwise. The TSR counterpart is detailed in Figure 11b.

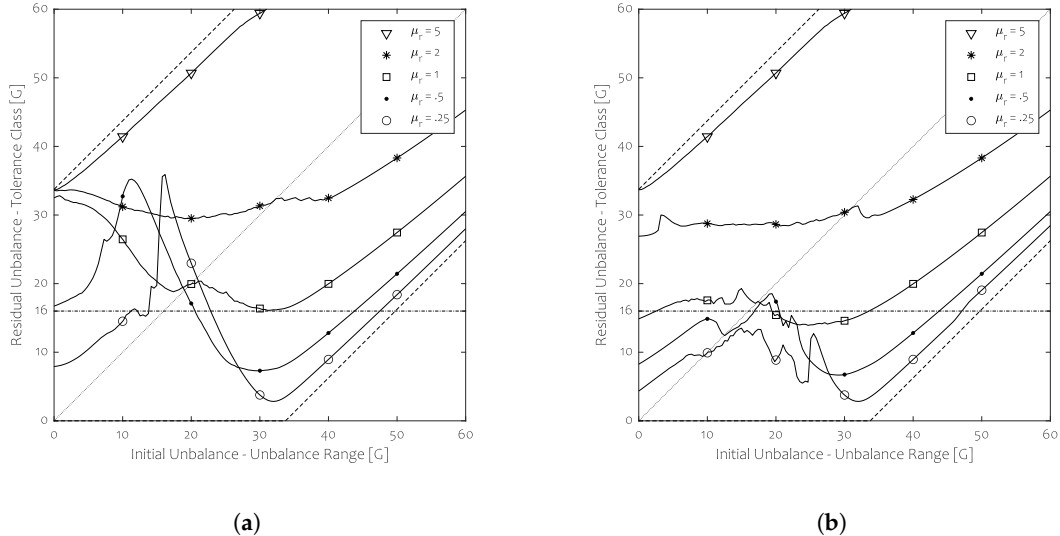

(**a**)　　　　　　　　　　　　　　　　　　　　　　　　　　　(**b**)

**Figure 11.** An increase of friction leads to a global deterioration of balancing capabilities for the STD and TSR technique. (**a**) STD, (**b**) TSR. Sweep-up and haziness of the balancing outcomes occur respectively for STD and TSR as stability is only guaranteed in a narrow upper part of the adopted speed profile for a low initial unbalance range.

The valid ranges in function of friction that can be attained are shown in Table 3 for the STD and TSR method respectively, assuming the *G*16 standard. The efficacy assessment shows the advantage of TSR as this method allows to operate more effectively. As raceway stiction is not easily affected, application of the TSR technique is a viable solution to enhance the efficacy of the ball balancer, provided that a variable speed drive is at hand. However, the lower speed plateau should be selected as such as to not compromise stable operation in the lower initial unbalance range.

We would like to recall as well that the way balancing capability is portrayed, such as in Figure 11a attempts to represent unbalance variability globally. This technique fails to discern any local behaviour, such as the gathering of balancing outcomes.

To conclude, reducing stiction is mainly favourable but its effect might not be prolific in the under-excited range as stability can be limited, both for the STD and TSR counterparts. Determining this stability limit lies outside the scope of this study.

**Table 3.** Efficacy assessment of both approaches for the G16 standard for various stiction parameters. Efficiency is defined as the ratio of valid unbalance range and total unbalance range for said tolerance class [*G*0–*G*50].

|  | STD | Eff. [%] | TSR | Eff. [%] |
|---|---|---|---|---|
| $\mu_r = 1$ | / | 0 | [*G*0–*G*3, *G*19–*G*34] | 36 |
| $\mu_r = 0.5$ | [*G*21–*G*43] | 44 | [*G*0–*G*17, *G*21–*G*44] | 80 |
| $\mu_r = 0.25$ | [*G*0–*G*14, *G*23–*G*47] | 76 | [*G*0–*G*47] | 94 |

The influence of damping on the attainable tolerance class is shown in Figure 12 for both run-up strategies. Determining the damping limit necessary for ensuring stable working conditions lies outside the scope of this study as well.

The valid ranges in function of system damping that can be attained are shown in Table 4 for the STD and TSR method respectively, assuming the *G*16 standard.

System damping ought to be chosen carefully (if applicable) as it restricts stability for the lower initial unbalance range. This can be seen in Figure 12a for low initial unbalance for $c_r = \{0.5; 0.6\}$ as these values entail a marked deterioration of balancing outcomes in the range [*G*0, *G*20].

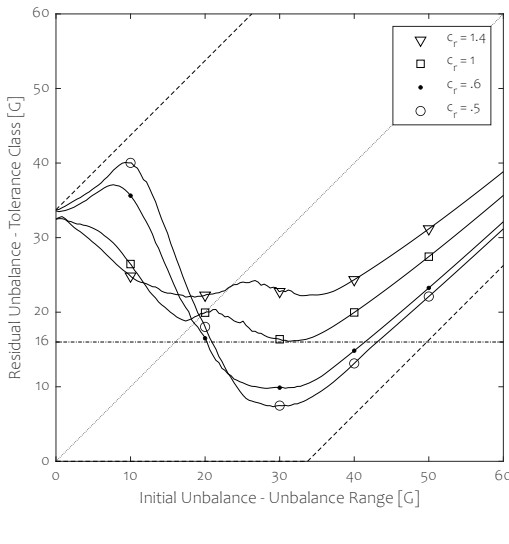
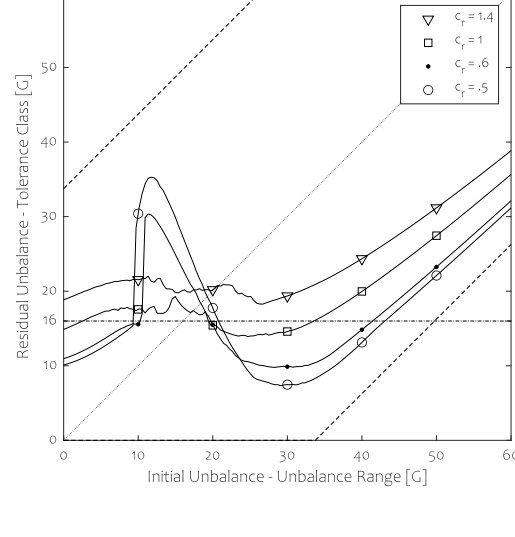

(**a**)                                                                 (**b**)

**Figure 12.** An increase of damping leads to a global deterioration of balancing capabilities for the STD and TSR technique. (**a**) STD, (**b**) TSR.

This section focused on the impact of stiction and damping on the complete initial unbalance range. Finding the most efficient combination of both parameters and applying these, in reality, is a daunting task but is outside the scope of this study. We wish to state again that TSR should be applied while ensuring that local stability is not void. The latter condition has an impact on the adopted temporary speed reduction as will be seen in the finalising subsection.

**Table 4.** Efficacy assessment of both approaches for the G16 standard for various damping parameters. Efficiency is defined as the ratio of valid unbalance range and total unbalance range for said tolerance class [*G0–G50*].

|  | STD | Eff. [%] | TSR | Eff. [%] |
|---|---|---|---|---|
| $c_r = 1$ | / | 0 | [*G0–G3, G19–G34*] | 36 |
| $c_r = 0.6$ | [*G20–G41*] | 42 | [*G0–G10, G20–G42*] | 64 |
| $c_r = 0.5$ | [*G21–G43*] | 44 | [*G0–G9, G21–G43*] | 62 |

### 4.6. Temporary Speed Reduction Plateau

As was mentioned previously, the lower speed of the TSR technique is a crucial parameter that should be matched to the global system stability operation boundaries. This value needs to be selected as such that

1. Effective unbalance reduction can occur to improve the balancing outcome of the STD approach;
2. Stable operation is guaranteed.

The range of lower speed plateau values that fulfil both requirements depends on the amount of initial unbalance as well, which is unfortunate since the initial unbalance is a priori unknown. Proper selection of the lower speed plateau remains therefore cumbersome. The valid speed range should, in practice, be selected by studying its influence on the complete scattering phenomenon in order to ensure the most effective approach.

The practical consequences of setting the lower speed plateau can be seen in Figure 13 for nominal damping and friction values ($\mu_r = c_r = 1$) and initial unbalance *G5*. In this graph, a standard run-up procedure was performed, up to nominal speed. Then, a gradual speed reduction occurred from nominal speed ($\Omega = 1.5\Omega_n$) to the undercritical speed $\Omega = 0.9\Omega_n$. The balancing outcomes are scattered for high speeds. Completing the TSR procedure occurs from right to left as the speed is decreased. Doing so leads to a prompt unbalance reduction near $\Omega = 1.1\Omega_n$. Unstable working occurs when nearing resonance which deteriorates some of the balancing outcomes. For illustration purposes, the lower speed plateau of the adopted speed profile ($\Omega = \frac{100\,\text{Hz}}{94.2\,\text{Hz}}\Omega_n = 1.06\Omega_n$) is denoted by the vertical dotted line. The operational boundary of the ball balancer is denoted as well by the horizontal dashed line. This boundary is attained by combining the initial unbalance (*G5*), and the total ball unbalance (*G34*). We assume that increasing the speed again does not influence the balancing outcome as doing so increases immobility, provided acceleration is not harsh. For completeness, a panoply of percentile values is provided, ranging from 5% to 95%.

From this graph, it is clear that an exaggerated speed reduction annihilates any improvement made so far. It should be noted from this graph that assessing the speed reduction limit by solely analysing the behaviour of the 95% percentile isoline is convenient as it takes into account all balancing outcomes. In order to simplify the upcoming discussion, only the 95% percentile isoline will be retained.

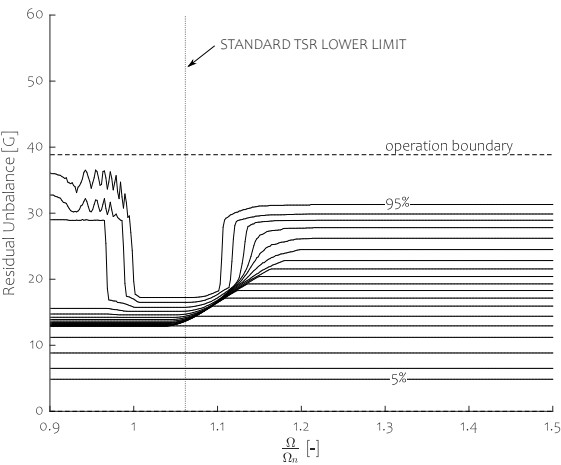

**Figure 13.** Effect of the lower speed plateau on the global balancing outcomes for $G_{ini} = 5$, $\mu_r = c_r = 1$.

Effect of Initial Unbalance on Valid Speed Plateau Range

The effect of the initial unbalance will be shown in the $[G0, G34]$ range as the TSR technique only influences this range. On Figure 14, the effect of the initial unbalance in combination with the selected lower speed plateau of the TSR technique is shown using a greyscale colour bar. The colour represents the residual unbalance attained in 95% of the cases, such as before. It serves as bound to all balancing outcomes. Darker shades denote higher residual unbalance bounds, while lighter shades denote a narrow and thus low-level bounding of the balancing outcomes. We can discern a pale mountain-like shape occurring near resonance, denoting where the TSR technique is effective. The TSR technique is only effective for low initial unbalance as an ABB operates efficiently for high initial unbalances. The left flank of the so-called mountain is grainy and dark as unstable behaviour prohibits steady operation. Only at a distance, far enough left or right of the mountain guarantees stable operation. We can discern in this Figure the three primary states of a ball balancer: Supercritical balancing behaviour, undercritical unbalancing behaviour aggravating initial unbalance and an unstable central transition around resonance.

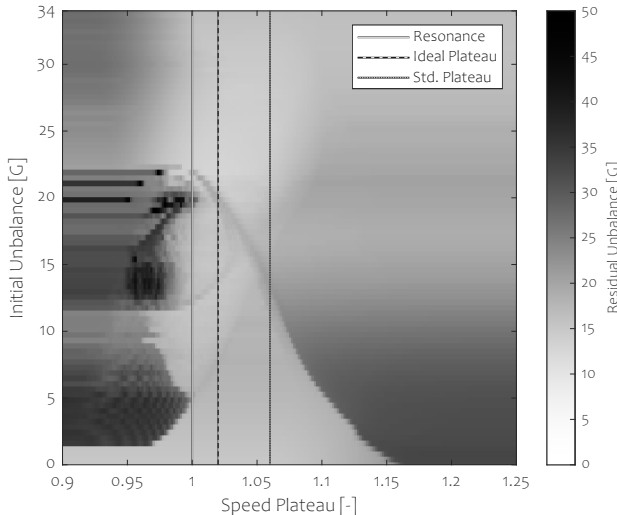

**Figure 14.** Effect of the initial unbalance on the valid lower speed plateau range, $\mu_r = c_r = 1$. The ideal speed plateau is 1.02.

As it is not possible to know the initial unbalance beforehand, it is interesting to look at the ideal lower speed plateau that guarantees for any initial unbalance on average the lowest spread of balancing outcomes. In this study, this initial unbalance range is $[G0, G34]$ as above this range,

TSR is no longer required. It is to be noted that by knowing a narrower initial unbalance range, one can benefit from this knowledge by fine-tuning the adopted speed plateau.

It makes sense now to posit that the rotational speed at which major relocation occurs depends on the initial unbalance. By analzing the average effect of the low speed plateau for under-exciting initial unbalance, i.e.,: $G_{ini} \leq G34$, we can state that the ideal plateau is 1.02 ( for $\mu_r = c_r = 1$). This is however not the rotational speed that was selected when presenting the former results of this study (the standard speed plateau is $\Omega = 1.06\Omega_n$). This higher speed was used as an attempt to ensure stable behaviour for different damping and friction values as well. This selection leads to the results presented in Figures 11 and 12. It is thus important to state that the lower plateau should be selected carefully as not to jeopardize stable operation while ensuring effective unbalance reduction. This ideal plateau depends on the amount of damping and stiction.

To finalize this section, a suitable lower speed value will now be selected for the worst conditions presented in Figure 12, for $\mu_r = 1, c_r = 0.5$ whereby the standard lower speed plateau was used. With these stiction and damping values, we discern in Figure 15 a similar mountain-like pattern. We discover that the lower speed plateau that was selected is too high as in the initial unbalance range of $[G10–G20]$, the relocation process did not occur yet. Further reducing the rotational speed up to $\Omega = 1.03\Omega_n$ reduces the unbalance outcomes in the initial range of $[G10–G20]$ considerably. However, extreme speed reduction will entail unstable operation in the initial unbalance range of $[G20–G25]$. This stability boundary prohibits further unbalance reduction. The ideal speed plateau, in this case, is 1.03. The resulting balancing outcome is presented in Figure 16 along with the STD outcome and the TSR results with the standard lower speed value of 1.06. Selecting the proper lower speed plateau affects the balancing behaviour profoundly. Spiking occurs near $G20$ for the ideal plateau selection. This happens as the response of the shaft to unbalance starts shifts due to the presence of the resonance. This effect shifts the preferred unbalance location towards a higher residual unbalance while still being stable. Further reducing speed will compromise global stability which is to be avoided at all costs.

It is clear by comparing Figures 14 and 15 that the shape of the effective TSR range can be altered. However, as the applicability of TSR relies on the system sensitivity, it is linked to the stable operation boundaries of the system as well. Care should thus be taken when tailoring the TSR technique for a specific rotordynamic setup.

This subsection can be concluded by stating that the effective range of TSR considerably depends on the lower speed plateau. This selection depends on the amount of damping, stiction and mainly the initial unbalance present. As this initial unbalance is in practice unknown, we have chosen to conduct our study as such to encompass this undetermined parameter as is. Knowing its variability will allow for adapting the lower speed plateau conveniently.

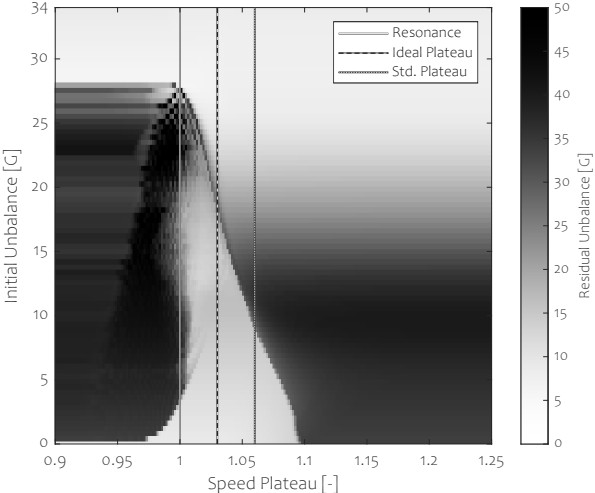

**Figure 15.** Effect of the initial unbalance on the valid lower speed plateau range, $\mu_r = 1, c_r = 0.5$. The ideal speed plateau is 1.03.

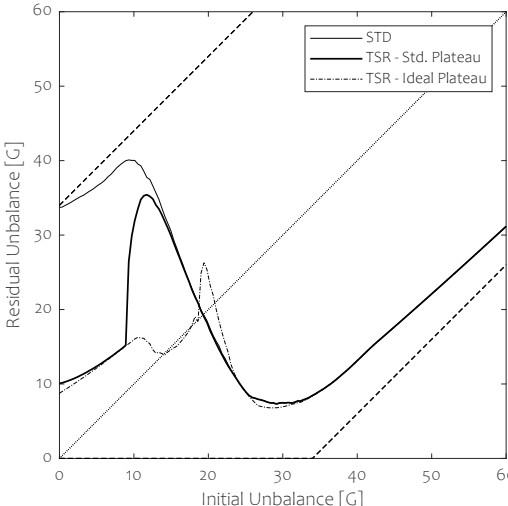

**Figure 16.** The balancing outcomes for $\mu_r = 1, c_r = 0.5$ whereby the lower speed plateau was altered from 1.06 (standard) to 1.03 (ideal). The STD balancing outcome is presented as well for illustration purposes.

## 5. Conclusions

This paper started by explaining how a 1-ball ABB mounted on a Jefcott rotor fundamentally works. Then, a rotordynamic model has been devised for a multi-ball ABB on a Jeffcott rotor using the Lagrangian approach, using relative cartesian coordinates. Rolling friction has been modelled hyperbolically. This model was used to perform a parametric study on the temporary speed reduction technique. The observations of this study summarise as follows:

1. Raceway friction has a significant impact on balancing scattering (Section 4.2)
2. An over-dimensioned ABB will balance poorly (Section 4.3).
3. TSR (temporary speed reduction) is an effective means for enhancing the balancing capabilities of an ABB.
4. The used speed profile severely affects the balancing outcome of a system, considering both overall balancing quality and transient resonant vibrations. TSR tackles this matter in a safe and consistent manner (Section 4.4).
5. System damping and friction should be evaluated carefully seen their complex impact on the lower unbalance range for a given tolerance class (Section 4.5).
6. The lower speed plateau of TSR should be selected for given damping and stiction values while taking into account the effect that the initial unbalance has on the stable operation conditions (Section 4.6).

We have shown that an ABB may be a peculiar balancing device due to its balancing scattering. However, the addition of this device is beneficial provided the system damping or rolling friction are contained. The provided tools allow for quantifying design parameters in order to facilitate the design procedure. We conclude that the TSR technique is a promising tool that should be considered when speed control is applicable.

**Author Contributions:** G.V.D.V.: conceptualization, methodology, investigation, validation, writing—original draft. B.V.: project administration, investigation, writing—review and editing. D.L.: supervision. P.G.: supervision, writing—review and editing. All authors have read and agreed to the published version of the manuscript.

**Funding:** The first author is a PhD Fellow of the Research Foundation Flanders—Fonds Wetenschappelijk Onderzoek. (FWO-1S08417N).

**Conflicts of Interest:** The authors declare no conflict of interest.

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
