# Peer review of "Let’s Make Ball Balancing Great Again: Why You Should Use Temporary Speed Reduction"

_machines, doi:10.3390/machines8040074_

Round 1

Reviewer 1 Report

The manuscript has to be considerably improved and cannot be considered for publication in this scientific journal in its present form.

Some of the issues that have to be considered by the authors are listed here below:

  • The Figure 1 should be explained in the manuscript text (2. Basic concept of an ABB) Please explain all the quantities shown in Figure also in the manuscript text.
  • The subsection 3.1. Equations of motion starts with the Figure 2. this Figure should be first explained in details. The text font in the Figure 2. is too small and not clear. Please rearrange the text and the Figure in this subsection. The same is valid for other subsections.
  • The Figure 3 should be placed in the subsection 4.1. Single run. Also please explain and discus in the manuscript text the quantity Vibration in micrometers shown in this graph.
  • The fonts in the figures should be enlarged to make the result data more clear to the readers.
  • Figures 15 and Figure 16 should not be in the Conclusions section.
  • The results shown in Figure 16. It seems that it contains 4 outcomes/curves, while the legend shows only 3. Please explain and/or correct.
  • Please check and correct the format of all references. For example, the reference 13 published by Lee, J. and Van Moorhem, W. is not correctly cited (journal name, volume pages are missing).

Some of the references contain the doi number – please be consistent.

In addition, more new references from 2017-2020 should be cited.

  • The title of the manuscript should be changed.

King regards

Reviewer 2 Report

A very interesting, narrowly specialized problem, nevertheless dealing with the solution of a well-known and current phenomenon. The authors analyze and describe the issue in detail, highly professionally. However, I would expand / supplement the used literature with several sources (also from other renowned magazines / publishers / the same is often repeated) dealing with the issue (diversify the list of references). I would describe the linked references in the text to references (eg [5-9], [13-17]), see where and what you used in your work / article. Advantages / disadvantages brief contribution of given resources. The article is well structured, logically structured, well-described results and understandable. The pictures are high quality, legible, they show the achieved results very nicely. A small comment on the location of the images so that they are closer to the text where they are described in the text. And at the end there should be no pictures. Non-standard form of closure, through reflectors.

I consider the article to be of high quality and I recommend it for publication after adding minor comments.

Reviewer 3 Report

The paper brings an in-depth analysis of a technique called Temporary Speed Reduction (TSR), previously presented by the authors, to further improve the performance of automatic ball balancers (ABB).

The paper is well written and organized. This is an interesting topic worth to be investigated and the method presented indeed seems to be a good improvement to the standard ABB. However, the title is a little presumptuous, the method presented has limitations as shown in the paper and is very sensitive to many parameters such as system damping and friction, which are difficult to identify.

Some graphics are a little hard to understand in a first glance.

Overall, it is a good paper a should be published after minor corrections.

Line 21: Typo "fundemantel".

Line 55: Typo "enduced".

Line 171: Typo "grade G a described by", shouldn't it be "grade G as described by".

Line 188: At the end of this sentece I would add "shows how severe the total amount of unbalance is (G grade)." for clarity.

Line 193: use lower case in the word "The" in "Nevertheless, The standard".

Figure 6: I recommend using some colors to make it easier to distinguish between the areas of the figure, colored lines could be used in the borders of the regions or light colors could be used in the background of each area (see figure attached). The description of table 2 is enough to define the areas, but using colors to mark the areas in the graph would improve the readability of the figure a lot.

Figure 13: x-axis label: use capital Omega instead of lower case omega (same for Omega_n).

In section 4.2, the conclusion I see is that when comparing the results between STD and TSR with the initial G20 unbalance, TSR reduces the residual unbalance in 98% of the cases and STD in 95% (line 227). But the real goal here is to achieve G16. And when comparing the results of STD and TSR, the STD achieves the goal in 85% of the cases and the TSR in 80% only (line 217). Comparing this with Figure 5, it seems that when the TSR works, it is typically better than STD, but the problem is that STD works in more cases than TSR (for the conditions evaluated - G16 goal with initial G20). A possible solution that I see from this analysis is that if the final balancing is bad, it is always possible to stop the machine and restart it again. This way the balls start in a different position and very likely a good balancing can be achieved in this next run. This is not desired, but it may be possible. It would be nice to include a discussion about this in the paper. Of course that this analysis is extended even further is section 4.3 when the initial unbalance is included in the discussion.

Section 4.4: The definition of immobility is not very clear (lines 284-285). How this value can assume values that are not integer numbers? Is this an average of the 1275 cases simulated?

I recommend using the work "position" instead of "stance" through the paper.

Reviewer 4 Report

In this paper is an in-depth study detailing how the technique should be implemented to guarantee effective balancing. By analysing a rotordynamic model of the Jeffcott kind, the influence of a multitude of parameters is studied such as the initial mass positions, the initial unbalance, the adopted speed profile, shaft damping, stiction and the speed reduction plateau of the adopted speed reduction strategy. The main findings of the study are that the adverse effects of stiction can be contained considerably using the speed reduction technique, especially in the under-excited range where a ball balancer behaves poorly when adopting a standard run-up profile. Finally, it should be stated that the speed plateau of the speed reduction technique should be selected carefully, preferably accounting for stiction, shaft damping and even more so the initial unbalance.
This paper has started by explaining how a 1-ball ABB mounted on a Jefcott rotor fundamentally works. Then a rotordynamic model has been devised for a multi-ball ABB on a Jeffcott rotor using the Lagrangian approach. This has been done using relative cartesian coordinates. Rolling friction has been modelled hyperbolically. This model was used to perform a parametric study on the temporary speed reduction technique.

In the paper the methods adequately described and the results are clearly presented. Good paper.

Main problems in editing and grammar:

In the page 14 there is an editing problem. Fill in the blanks on the page.

In rows:

21 fundamental
32 plug-and-play device
55 induced
63 assess
Table 2. Occurrence
308 techniques
310 indicates
320 the lowest
Figure 11. guaranteed

and there are many little grammar mistakes, so grammar check is highly required.

Please check and correct the format of all references, too.

Round 2

Reviewer 1 Report

The manuscript has been significantly improved.

I suggest only the following minor correction:

The word 'state' in the last sentence of the Abstract should be changed into 'stated':

'Finally, it should be state that the speed plateau'

Author Response

Dear,

Thank you for your comment. The segment ''it should be state that" was highlighted in red in the article with highlighted differences to denote that this segment was removed. In the last version, the complete sentence reads 'Finally, the speed plateau of the speed reduction technique should be selected carefully, preferably accounting for stiction, shaft damping and even more so the initial unbalance.'

Regards,

Gabriël Van De Velde

Reviewer 3 Report

All the suggestions were addressed satisfactorily.

Author Response

This reply is not necessary since there are no comments.

Regards, Gabriël Van De Velde

Reviewer 4 Report

Most of the problems were checked and corrected

page 10 of 20 - Table 2. Occurrence
page 10 of 20 - Figure 11. guaranteed

226 ... both the standard (STD) and TSR procedures!!! are ...
334 ... both the STD and TSR techniques!!! are ...
372 ... both for the STD and TSR counterparts!!!.
"Both" cannot be used with a singular of nouns, as it always refers to two things.
Like You uses it in other sentences:
both of the mentioned issues
both frames
both runs
both balancing masses
both methods
both procedures

Or STD is neither a procedure nor a technique?

242 ...of the STD and TSR methods!!! is respectively 95% and 98%. This means that in the majority of cases both methods achieve a balancing class smaller than the initial G20.

Author Response

Dear,

All comments have been accepted and implemented. It seems that some artefacts of the Dutch language have affected my writing skills (concerning the 'both'-issue).

We thank you for this comment.

Kind regards,

Gabriël Van De Velde